# Study on Rock Failure Criterion Based on Elastic Strain Energy Density

**Yang Cheng [1],* and Liangliang Zhang [2]**

1    College of Architectural Engineering, Tongling University, Tongling 244000, China
2    School of Civil Engineering and Architecture, Anhui University of Science and Technology, Huainan 232001, China; zllaust@163.com
*    Correspondence: chengytl@163.com

**Abstract:** Uniaxial and five conventional triaxial compression tests were conducted on sandstone to obtain the evolution laws of the input energy density, elastic strain energy density, and dissipative energy density. The input and dissipative energy densities increased with increasing axial strain; the elastic strain energy density increased with increasing axial strain at the pre-peak stage and decreased after the peak. According to the linear change rule between the peak elastic strain energy density and confining pressure, the energy density failure criterion of sandstone was established, and the criterion has high precision and few parameters, and the parameters have clear physical meaning. Moreover, the expression of the energy density failure criterion was similar to the classical Hoek-Brown criterion, but its adaptability was more extensive. The strength calculation results for seven different rocks under different confining pressures calculated using the energy density failure criterion were consistent with the experimental values, and the calculation error was smaller than that of the Mohr–Coulomb criterion and Drucker–Prager criterion, verifying the accuracy and applicability of the criterion.

**Keywords:** sandstone; elastic strain energy; dissipative energy; failure criteria





## 1. Introduction

The rock strength theory and deformation failure characteristics form the theoretical basis for research and analysis of rock engineering safety and stability. For years, rock strength and failure criteria based on classical elastic–plastic theory have been the basis for judging engineering failure [1]. However, rocks are typically nonuniform and discontinuous media that contain a large number of structurally discontinuous and irregularly shaped microcracks. The failure mechanism of rocks is closely related to the mode of action of the host environment, and the mechanical properties of rocks at failure show complex changes under different loading forms. Some strength theories and failure criteria make it difficult to effectively analyze the complex strength changes and failure behaviors of rocks. The thermodynamic theory states that material failure is a state instability phenomenon driven by energy, and the transfer of energy is the essence of material deformation and failure [2]. Therefore, studying and establishing the energy variation pattern during the rock failure process and its relationship with the strength and overall failure from an energy perspective will be more conducive to reflecting the essential characteristics of rock strength variation and overall failure under external loads.

Recently, domestic and international scholars have conducted studies on the law of energy evolution during rock deformation. Liu et al. [3] conducted uniaxial cyclic loading tests on sandy mudstones and siltstones, obtained damage variables based on the dissipated energy, and established a damage constitutive model that could describe the mechanical properties of rocks under cyclic loading according to the Lemaitre strain equivalence principle. Xie et al. [4] showed that the dissipated energy generated under an external

load damages the rock and reduces its strength and that the failure of the rock is caused by the sudden release of the internally accumulated strain energy. Lia et al. [5] conducted conventional triaxial tests on fine-to-medium-grained granite under different confining pressures and loading and unloading stress paths to study their energy evolution and failure processes. Bagde and Petroš [6] found that the energy required to cause fractures increased rapidly with increasing amplitude and frequency in dynamic cyclic loading. Fuenkajorn et al. [7] demonstrated that under different loading rates, the distortion energy during rock expansion and failure follows a linear relationship with the average normal stress. Peng et al. [8] experimentally studied the energy dissipation and release during coal failure under conventional triaxial compression and proposed two parameters (failure energy ratio and stress drop coefficient) to describe the failure mode of coal under different confining pressures. Song et al. [9] studied the damage and failure of coal and rock with bursting liability from the perspective of energy dissipation using the geophysical method of electromagnetic radiation and explored the energy dissipation characteristics during uniaxial compression and their main influencing factors. Zhou et al. [10] established an energy mutation failure criterion based on the evolution law of gravitational potential energy, elastic strain energy, and kinetic energy. The criterion has the advantages of clear physical meaning, strong integrity, and good applicability y, which can accurately predict the instability and failure of jointed slopes. Wang et al. [11] conducted conventional triaxial compression tests on marbles and sandstones with different bedding plane dip angles under different confining pressures, obtained the influences of confining pressure on critical strain energy release rate and energy transformation of anisotropic intact rocks, and established a new critical strain energy release rate failure criterion. Gennady et al. [12] use the energy approach in the mathematical modeling of mechanical systems; the fracture criterion does not require integration to calculate the strain energy and dissipation energy. The aforementioned studies primarily analyzed the energy types and evolution laws during the deformation and failure of rocks under different conditions of stress; However, most of the existing research establishes rock failure criteria from the mechanical point of view, and few studies establish rock failure criteria from the relationship between peak elastic strain energy density and confining pressure.

In this paper, sandstone is taken as the research object. Firstly, uniaxial and five sets of conventional triaxial compression tests under different confining pressures were conducted to study the evolution law of input energy density, elastic strain energy density, and dissipative energy density of sandstone during deformation and failure. Secondly, according to the test results, the fitting relationship between the peak elastic strain energy density and the confining pressure was obtained, and combined with the theoretical calculation formula of the peak elastic strain energy density, the rock energy density failure criterion was established. Finally, experimental data from seven different types of rocks were used to verify the rationality and accuracy of the energy density failure criterion.

## 2. Conventional Triaxial Compression Test of Sandstone

### 2.1. Test Method

A ZTCR-2000 low-temperature rock triaxial system produced in Changchun City, Jilin Province, China (Figure 1) was used to conduct uniaxial compression and five sets of conventional triaxial compression tests on sandstone. The test system provided a maximum confining pressure of 50 MPa and a maximum axial force of 2000 kN, which satisfied the strength requirements of this test. To avoid contingencies in the stress and strain data of the samples, three samples were used for each group of confining pressure tests, and the optimal stress and strain curves were obtained according to the test results. Sandstone was taken from the roof bedrock of the No. 8 coal seam in Banji Coal Mine, Lixin County, Anhui Province, China, with a buried depth of 619 m~652 m. All samples prepared for tests have a diameter of 50 mm and a height of about 100 mm. During the experiment, the sandstone sample was first pre-loaded axially at a loading rate of 100 N/s up to 1.0 MPa and then subjected to a confining pressure at the predetermined values of 5, 10, 15,

20, and 25 MPa and a loading rate of 200 N/s. After the confining pressure reached a predetermined value, it was stabilized for 10 s. Finally, an axial pressure was applied at a loading rate of 0.06 mm/min until the sandstone sample was damaged. After the sample was damaged, the axial pressure was unloaded, and a confining pressure was applied. The sample was removed, and photographs were taken to save and export the test data.

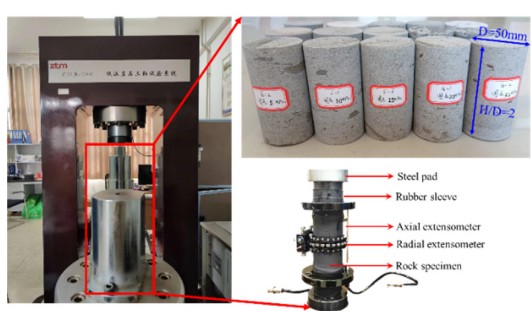

**Figure 1.** Sandstone specimens and test system.

### 2.2. Analysis of Test Results

The stress–strain curves of the sandstone samples obtained from the uniaxial and triaxial compression tests are shown in Figure 2.

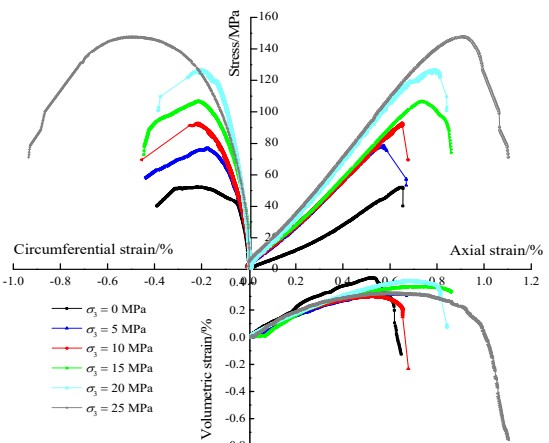

**Figure 2.** Stress–strain curve.

The basic mechanical parameters of the sandstone obtained from the experimental results are listed in Table 1, and the cohesion (*c*) and internal friction angle ($\varphi$) are obtained according to the Mohr–Coulomb criterion. The relationships between the confining pressure and the sandstone peak strength, peak axial strain, peak circumferential strain, peak volumetric strain, elastic modulus, and Poisson's ratio are shown in Figure 3.

**Table 1.** Mechanical parameters of sandstone.

| $\sigma_3$/MPa | $\sigma_c$/MPa | Peak Axial Strain/% | Peak Circumferential Strain/% | Peak Volumetric Strain/% | E/GPa | $\mu$ | c/MPa | $\varphi$/(°) |
|---|---|---|---|---|---|---|---|---|
| 0 | 52.32 | 0.56 | −0.22 | 0.12 | 9.33 | 0.18 | | |
| 5 | 78.53 | 0.57 | −0.18 | 0.21 | 14.94 | 0.20 | | |
| 10 | 92.70 | 0.65 | −0.22 | 0.21 | 15.02 | 0.21 | 14.44 | 34.75 |
| 15 | 106.88 | 0.74 | −0.19 | 0.36 | 16.19 | 0.19 | | |
| 20 | 130.68 | 0.79 | −0.21 | 0.37 | 18.06 | 0.19 | | |
| 25 | 147.81 | 0.91 | −0.51 | −0.11 | 18.62 | 0.21 | | |

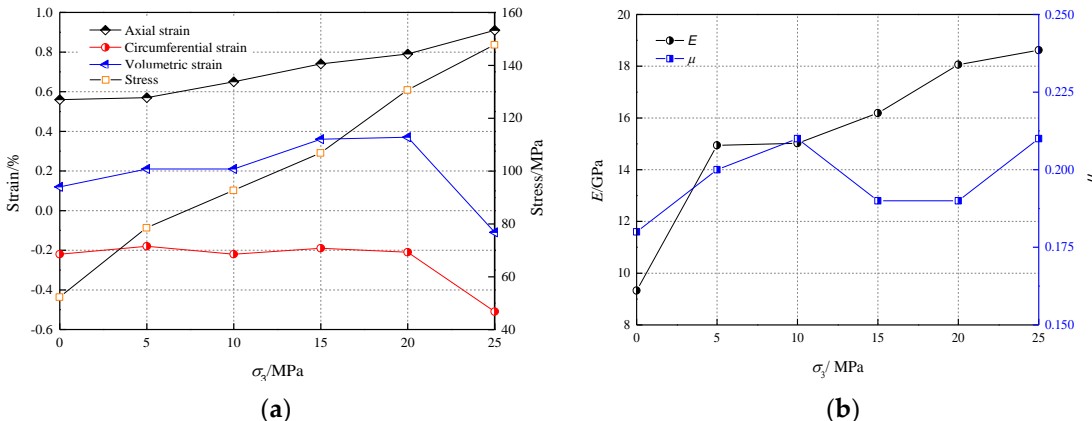

(**a**)  (**b**)

**Figure 3.** Relationship between mechanical parameters and confining pressure: (**a**) axial stress and strain, (**b**) elastic modulus and Poisson's ratio.

Table 1 and Figure 3 show that the peak strength, peak axial strain, and elastic modulus of sandstone increase significantly with an increase in the confining pressure. When the confining pressure increases from 0 to 25 MPa, the peak strength, peak axial strain, and elastic modulus increase by 182.5%, 62.5%, and 99.6%, respectively, indicating that confining pressure can effectively improve the strength and deformation resistance of sandstone. When the confining pressure is between 0–20 MPa, the peak circumferential and volumetric strains of the sandstone change slightly, whereas when the confining pressure is 25 MPa, both significantly decrease. This is because, under high confining pressure, the side of the sandstone sample is greatly restricted, resulting in a smaller circumferential and volumetric deformation during its failure. As the confining pressure increases, the Poisson's ratio fluctuates between 0.18 and 0.21, indicating a small correlation between the two.

The failure modes of the sandstone under different confining pressures are shown in Figure 4. As shown in the figure, the failure mode of the sandstone varied under different confining pressures. When the confining pressures are 0 and 5 MPa, the failure mode is vertical compression failure, and the failure surface is parallel to the axial direction of the sandstone; when the confining pressures are 10, 15, 20, and 25 MPa, the failure mode is an oblique shear failure. This is because a higher confining pressure suppresses the expansion of vertical cracks, and the shear stress on the inclined section of sandstone under a triaxial high-stress state is greater than its shear strength, ultimately leading to oblique shear failure along the weak plane.

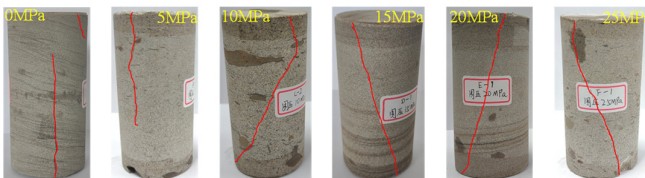

**Figure 4.** Failure modes of sandstone.

Figure 4 also shows that when the confining pressures are 0 and 5 MPa, the fracture angle of the sandstone is approximately 90°, and under other confining conditions, it is approximately 45°. This indicates that the fracture angle of sandstone under high confining pressure was smaller than that under low confining pressure. This is because when the confining pressure is high, sandstone failure requires a greater load, and the internal friction effect of sandstone is more evident, which suppresses the fracture of sandstone and reduces the fracture angle.

## 3. Sandstone Energy Analysis

### 3.1. Theoretical Analysis

According to thermodynamic theory, the input energy of the system is equal to the sum of the elastic strain energy and energy dissipated, regardless of the impact of the external temperature change and material exchange on the test system during the test. The elastic strain energy is the energy accumulated inside a sandstone sample when it undergoes elastic deformation. The dissipated energy primarily included the plastic strain energy of the sandstone sample when it underwent plastic deformation, and the surface energy and various radiation energies were consumed during the development and penetration. According to the principle of energy conservation, the input, elastic strain, and dissipation energies satisfy the following equation [13]:

$$VU_F = V(U_E + U_D) \tag{1}$$

where $V = \frac{\pi}{4}D^2H$ is the volume of the sandstone sample; $D$ and $H$ are the diameter and height of the sandstone sample, respectively; $U_F$, $U_E$, and $U_D$ are input energy, elastic strain energy, and dissipative energy densities, respectively.

During the conventional triaxial compression test, the input energy density of the experimental system for sandstone samples can be expressed as [14]

$$U_F = \int_0^{\varepsilon_1} \sigma_1 d\varepsilon_1 + 2\int_0^{\varepsilon_3} \sigma_3 d\varepsilon_3 \tag{2}$$

where $\sigma_1$ and $\sigma_3$ are the maximum and minimum principal stresses, and $\varepsilon_1$ and $\varepsilon_3$ are the principal strains in the direction of the maximum and minimum principal stresses, respectively.

According to the theory of elasticity, the elastic strain energy density is [15]

$$U_E = \frac{1}{2}(\sigma_1 \varepsilon_1^{\rm e} + 2\sigma_3 \varepsilon_3^{\rm e}) = \frac{1}{2E}\left[\sigma_1^2 + 2(1-\mu)\sigma_3^2 - 4\mu\sigma_1\sigma_3\right] \tag{3}$$

By substituting Equations (2) and (3) into Equation (1), the dissipative energy density can be obtained as follows:

$$U_D = \int_0^{\varepsilon_1} \sigma_1 d\varepsilon_1 + 2\int_0^{\varepsilon_3} \sigma_3 d\varepsilon_3 - \frac{1}{2E}\left[\sigma_1^2 + 2(1-\mu)\sigma_3^2 - 4\mu\sigma_1\sigma_3\right] \tag{4}$$

where $E$ is elastic modulus; $\mu$ is Poisson's ratio.

The formulae for calculating the input energy, elastic strain energy, and dissipative energy densities are as follows:

$$\begin{cases} U_F = \int_0^{\varepsilon_1} \sigma_1 d\varepsilon_1 + 2\int_0^{\varepsilon_3} \sigma_3 d\varepsilon_3 \\[2mm] U_E = \dfrac{1}{2E}\left[\sigma_1^2 + 2(1-\mu)\sigma_3^2 - 4\mu\sigma_1\sigma_3\right] \\[2mm] U_D = \int_0^{\varepsilon_1} \sigma_1 d\varepsilon_1 + 2\int_0^{\varepsilon_3} \sigma_3 d\varepsilon_3 - \dfrac{1}{2E}\left[\sigma_1^2 + 2(1-\mu)\sigma_3^2 - 4\mu\sigma_1\sigma_3\right] \end{cases} \tag{5}$$

### 3.2. Energy Density Analysis

Based on the conventional triaxial compression test results of sandstone under different confining pressures, the change curves of the input energy, elastic strain energy, and dissipative energy densities of the sandstone with the axial strain were obtained according to Equation (5) and the results are shown in Figure 5.

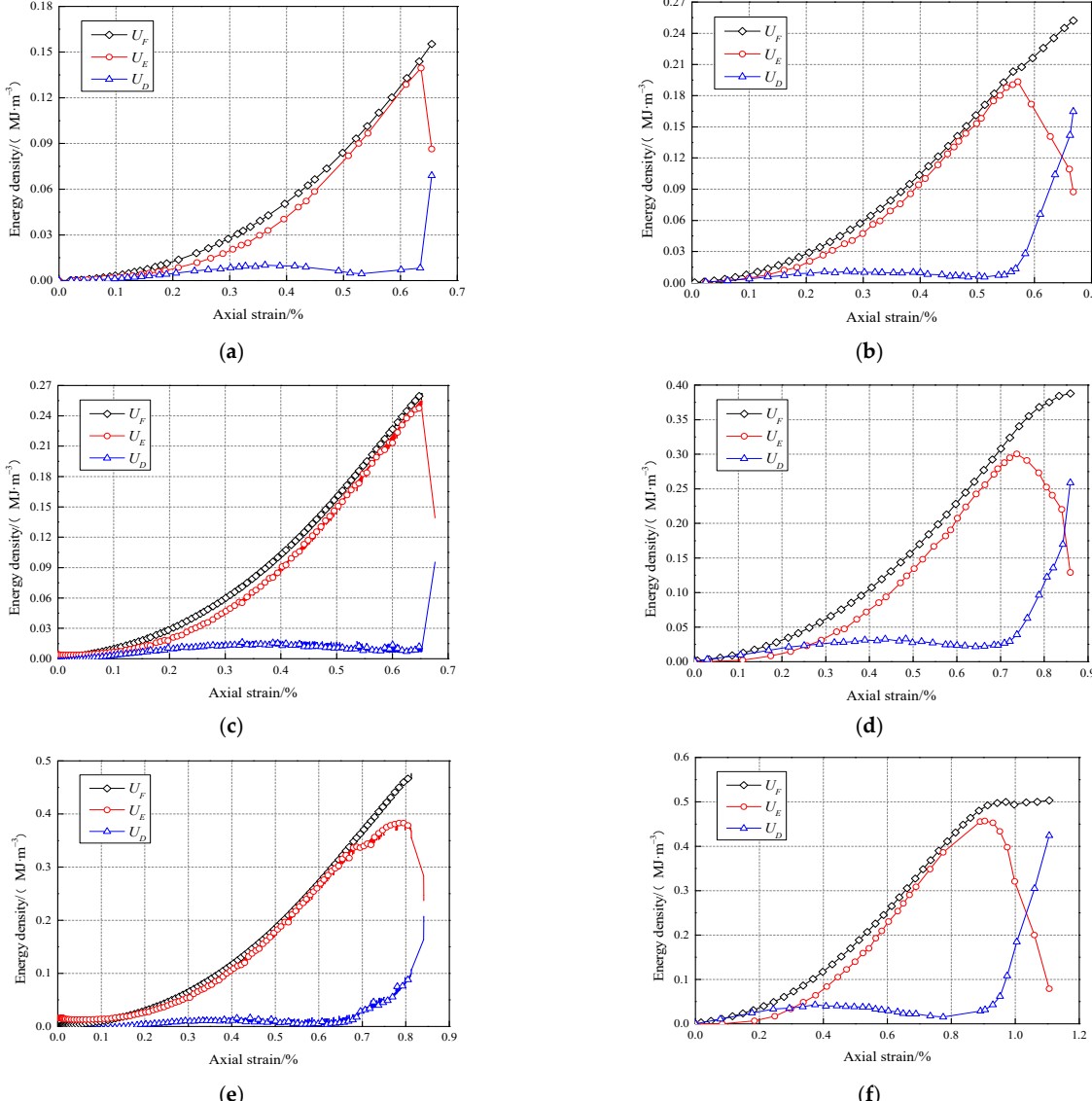

**Figure 5.** Relationship between energy density and axial strain. (**a**) $\sigma_3$ = 0 MPa. (**b**) $\sigma_3$ = 5 MPa. (**c**) $\sigma_3$ = 10 MPa. (**d**) $\sigma_3$ = 15 MPa. (**e**) $\sigma_3$ = 20 MPa. (**f**) $\sigma_3$ = 25 MPa.

It can be observed from Figure 5 that under different confining pressures, the trends of changes in the sandstone input energy, elastic strain energy, and dissipative energy densities with the axial strain are the same. The input energy density always increased throughout the deformation and failure processes of the sandstone because, throughout the experimental process, the system always loaded the sandstone and continuously input energy into it. The elastic strain energy density increased gradually in the pre-peak stage, and the value was close to the input energy density, which indicates that the pre-peak input energy transformed into elastic strain energy and was stored in the sample. After the sandstone underwent failure, the elastic strain energy density decreased sharply, and the elastic strain energy stored before the peak was released rapidly. The dissipative energy density was extremely low in the pre-peak stage, whereas it increased sharply in the post-peak stage. The energy density curve shows that the accumulation of elastic strain energy was the main factor before the deformation and failure of sandstone, and this component of the energy was the driving force for the failure of the samples. Each sample had an elastic strain energy storage limit; when the amount of input energy that was converted into elastic strain energy exceeded this limit, the sample could no longer

store energy and was damaged. Energy dissipation is the main form of sandstone failure. Most of the input energy of the external force was dissipated rapidly by crack initiation, propagation, coalescence, and friction on the fracture surface, resulting in a sharp increase in the post-peak dissipated energy density with strain and a continuous decrease in the elastic strain energy density. From the perspective of mechanics, the failure of sandstone was caused by the external load exceeding the compressive strength of the sample itself, while from the perspective of energy, it was caused by the fact that the elastic strain energy accumulated in the test system inside the sandstone sample exceeded its storage limit.

The relationship between the confining pressure and peak input energy density $U_{FP}$ is shown in Figure 6. As shown in the figure, the peak input energy density $U_{FP}$ increased with increasing confining pressure. When the confining pressure was 0, 5, 10, 15, 20, and 25 MPa, $U_{FP}$ increased from 0.1507, 0.2081, 0.2619, 0.3364, and 0.4517 MJ/m$^3$ to 0.4917 MJ/m$^3$, respectively. The increases were 38.81%, 25.85%, 28.44%, 34.27%, and 8.86%, respectively. The $U_{FP}$ under a confining pressure of 25 MPa was 3.26 times that under uniaxial compression. This is because the larger the confining pressure is, the stronger the ability of sandstone to resist damage and deformation; thus, more energy must be absorbed when a failure occurs.

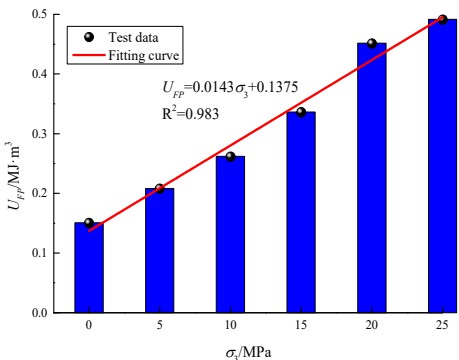

**Figure 6.** Relationship between confining pressure and $U_{FP}$.

## 4. Sandstone Failure Criterion

The relationship curve between the confining pressure and peak elastic strain energy density $U_{EP}$ obtained from the test results is shown in Figure 7.

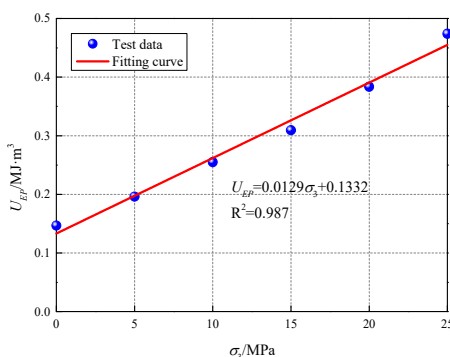

**Figure 7.** Relationship between confining pressure and $U_{EP}$. of sandstone.

The peak elastic strain energy density of sandstone under conventional triaxial compression increases with an increase in the confining pressure and exhibits a good linear relationship where $R^2$ is 0.987, indicating that the confining pressure can effectively improve the energy storage limit of sandstone. To further verify the generality of this rule, the peak elastic strain energy densities $U_{EP}$ of seven different types of rocks under conventional triaxial compression failure were statistically collected, and the results are presented in Table 2.

**Table 2.** Peak elastic energy density of seven types of rocks.

| References | Rock Types | $\sigma_3$/MPa | $\sigma_c$/MPa | $U_{EP}$/(MJ·m$^{-3}$) |
|---|---|---|---|---|
| [16] | Chlorite schist | 0 | 34.57 | 0.0408 |
| | | 5 | 51.51 | 0.0608 |
| | | 20 | 78.67 | 0.1062 |
| | | 30 | 97.36 | 0.1539 |
| | | 40 | 115.11 | 0.2138 |
| [17] | Hengda sandstone | 0 | 76.07 | 0.1027 |
| | | 10 | 108.23 | 0.1796 |
| | | 20 | 132.95 | 0.2439 |
| | | 30 | 148.19 | 0.2759 |
| | | 40 | 169.31 | 0.3324 |
| [18] | Argillaceous dolomite | 0 | 20.20 | 0.0358 |
| | | 5 | 67.80 | 0.2155 |
| | | 10 | 107.50 | 0.3694 |
| | | 15 | 132.00 | 0.4792 |
| | | 20 | 171.00 | 0.6431 |
| [19] | Jinping greenschist | 0 | 39.87 | 0.0757 |
| | | 4 | 65.63 | 0.1677 |
| | | 8 | 88.79 | 0.2041 |
| | | 20 | 133.92 | 0.3499 |
| | | 40 | 173.03 | 0.4385 |
| | | 50 | 188.68 | 0.5577 |
| [20] | Huashan granite | 0 | 140.36 | 0.2143 |
| | | 15 | 272.36 | 0.5994 |
| | | 25 | 313.82 | 0.7263 |
| | | 35 | 376.00 | 0.9854 |
| [21] | Jinping marble | 0 | 82.31 | 0.1831 |
| | | 5 | 146.12 | 0.3466 |
| | | 10 | 187.21 | 0.5091 |
| | | 20 | 246.34 | 0.8211 |
| | | 30 | 290.78 | 1.0484 |
| | | 40 | 331.74 | 1.0899 |
| [22] | Granodiorite | 0 | 164.62 | 0.2596 |
| | | 1 | 176.79 | 0.3025 |
| | | 2.5 | 203.88 | 0.4043 |
| | | 5 | 217.59 | 0.4418 |
| | | 10 | 289.88 | 0.7263 |
| | | 15 | 310.43 | 0.8377 |
| | | 20 | 347.14 | 1.0591 |

Based on the data in Table 2, the curves of the confining pressure and peak elastic strain energy density of the different rock types are shown in Figure 8.

As shown in Figure 8, the peak elastic strain energy density and confining pressure of the seven different types of rocks satisfy the linear growth relationship, and the fitting formula is as follows:

$$U_{EP} = a + b\sigma_3 \tag{6}$$

where *a* represents the peak elastic strain energy density of the rock under uniaxial compression. Combined with the boundary condition, $\sigma_1|_{\sigma_3=0} = \sigma_c$, $a = \dfrac{\sigma_c^2}{2E}$, and $\sigma_c$ represent the uniaxial compressive strength of the rock, and *b* is the slope of the peak elastic strain energy density of the rock with increasing confining pressure.

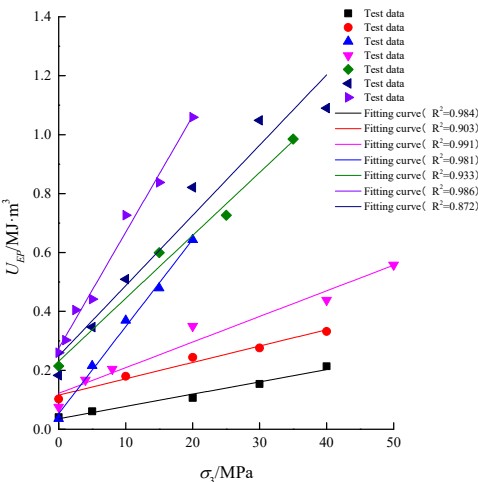

**Figure 8.** Relationship between confining pressure and $U_{EP}$ of of seven different types of rocks.

During the compression of rock samples, the elastic strain energy gradually increases with an increase in the external force. However, because the energy input by the system gradually transforms into dissipated energy with the plastic deformation of the samples, the elastic strain energy cannot continue to increase; that is, there is an elastic strain energy storage limit for rock samples. When the elastic strain energy reaches the storage limit, the sample is damaged. According to Equation (3), the storage limit of the rock elastic strain energy $U_{E\max}$ is

$$U_{E\max} = \frac{1}{2E}\left[\sigma_1^2 + 2(1-\mu)\sigma_3^2 - 4\mu\sigma_1\sigma_3\right] \tag{7}$$

The elastic strain energy density corresponding to the peak stress of the rock sample is $U_{EP}$. When the sample is damaged, the elastic strain energy density satisfies

$$U_{E\max} = U_{EP} \tag{8}$$

The combination of Equations (6)–(8) is

$$\frac{1}{2E}\left[\sigma_1^2 + 2(1-\mu)\sigma_3^2 - 4\mu\sigma_1\sigma_3\right] = b\sigma_3 + \frac{\sigma_c^2}{2E} \tag{9}$$

The rock failure criterion based on elastic strain energy density (hereinafter referred to as "E–D failure criterion") obtained by simplifying Equation (9) is

$$\sigma_1 = 2\mu\sigma_3 + \sqrt{(4\mu^2 + 2\mu - 2)\sigma_3^2 + 2E\left(b\sigma_3 + \frac{\sigma_c^2}{2E}\right)} \tag{10}$$

Assuming that the volumetric strain during rock failure is zero [23], Equation (10) can be further simplified as

$$\sigma_1 = \sigma_3 + \sqrt{2Eb\sigma_3 + \sigma_c^2} \tag{11}$$

Equation (11) is the rock energy density failure criterion, which has a simple form and contains fewer parameters. The physical meaning of each parameter is clear. Further observation reveals that the form of this failure criterion is similar to the expression of the classic Hoek–Brown failure criterion [24], as follows:

$$\sigma_1 = \sigma_3 + \sqrt{m\sigma_c\sigma_3 + \sigma_c^2} \tag{12}$$

where $m$ denotes the Hoek–Brown failure criterion. When $b = \frac{m\sigma_c}{2E}$, the energy density failure criterion can be transformed into the Hoek–Brown failure criterion such that the Hoek–Brown failure criterion is a special case of the energy density failure criterion. How-

ever, the use of the Hoek–Brown failure criterion has strict limitations, as its third principal stress should be less than half of the uniaxial compressive strength. The energy density failure criterion is derived theoretically based on the results of triaxial compression tests, and its application is not limited by the third principal stress. Therefore, compared with the Hoek–Brown failure criterion, its application range is wider.

The accuracy of the energy density failure criterion was verified using sandstone triaxial compression test data, as shown in Figure 9.

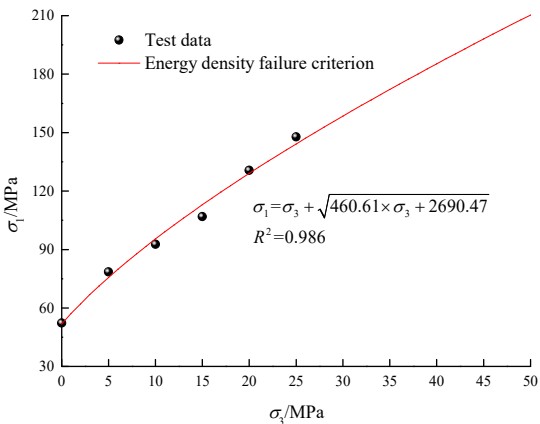

The equation shown in the figure is:

$$\sigma_1 = \sigma_3 + \sqrt{460.61 \times \sigma_3 + 2690.47}$$
$$R^2 = 0.986$$

**Figure 9.** Comparison between calculated results and test results.

From the figure, it can be observed that the energy density failure criterion calculates the strength of sandstone, and the experimental results are consistent, with $R^2 = 0.986$, thus verifying the accuracy of the energy density failure criterion.

## 5. Discussions

To further investigate the feasibility and applicability of the energy density failure criterion, experimental results of seven types of rocks were used to verify the energy density failure criterion and compared with experimental values, such as the Mohr–Coulomb criterion (M–C) and Drucker–Prager criterion(D–P) calculation values. The results are presented in Table 3 and Figure 10.

Figure 10 shows that the seven different rock strengths calculated by the D–P criterion are all lower than the experimental values, and their accuracy is the lowest compared to the M–C criterion and E–D criterion. Although the accuracy of the M–C criterion is higher than that of the D–P criterion, the relationship between the rock strength calculated by this criterion and the confining pressure is linear. The seven different rock test results showed that the relationship between the rock compressive strength and confining pressure is nonlinear. Therefore, when the confining pressure was high, the difference between the rock strength calculated using the M–C criterion and the test value was significant. The relationship between the compressive strength of the seven types of rocks calculated by the E–D criterion and the confining pressure was nonlinear. The calculated results were consistent with the test results, and the accuracy was higher than that of the M–C and D–P criteria. Therefore, the E–D criterion established in this study, based on the elastic strain energy, can be used to calculate the compressive strength of different types of rocks under different confining pressure conditions, and its feasibility and accuracy were further verified.

**Table 3.** Comparison results of the strengths of seven different rocks.

| Rock Type | $\sigma_3$/MPa | Test Value/MPa | Calculated Value/MPa | | |
|---|---|---|---|---|---|
| | | | M-C | D-P | E-D |
| Chlorite schist | 0 | 34.57 | 38.31 | 29.36 | 34.57 |
| | 5 | 51.51 | 48.06 | 37.89 | 46.97 |
| | 20 | 78.67 | 77.31 | 63.48 | 78.83 |
| | 30 | 97.36 | 96.81 | 80.54 | 97.78 |
| | 40 | 115.11 | 116.31 | 97.60 | 115.68 |
| Hengda sandstone | 0 | 76.07 | 81.66 | 60.66 | 76.07 |
| | 10 | 108.23 | 104.26 | 80.26 | 103.32 |
| | 20 | 132.95 | 126.86 | 99.87 | 127.84 |
| | 30 | 148.19 | 149.46 | 119.47 | 150.63 |
| | 40 | 169.31 | 172.06 | 139.07 | 172.18 |
| Argillaceous dolomite | 0 | 20.20 | 26.54 | 11.53 | 20.2 |
| | 5 | 67.80 | 63.14 | 30.66 | 77.29 |
| | 10 | 107.50 | 99.74 | 49.78 | 110.21 |
| | 15 | 132.00 | 136.34 | 68.91 | 136.90 |
| | 20 | 171.00 | 172.94 | 88.03 | 160.27 |
| Jinping greenschist | 0 | 39.87 | 57.08 | 38.74 | 39.87 |
| | 4 | 65.63 | 68.48 | 47.75 | 61.09 |
| | 8 | 88.79 | 79.88 | 56.75 | 78.20 |
| | 20 | 133.92 | 114.08 | 83.78 | 119.69 |
| | 40 | 173.03 | 171.08 | 128.82 | 175.22 |
| | 50 | 188.68 | 199.58 | 151.34 | 199.86 |
| Huashan granite | 0 | 140.36 | 151.76 | 70.31 | 140.36 |
| | 15 | 272.36 | 250.91 | 124.84 | 261.79 |
| | 25 | 313.82 | 317.01 | 161.20 | 322.27 |
| | 35 | 376.00 | 383.11 | 197.55 | 375.35 |
| Jinping marble | 0 | 82.31 | 110.28 | 54.20 | 82.31 |
| | 5 | 146.12 | 139.93 | 71.49 | 136.30 |
| | 10 | 187.21 | 169.58 | 88.78 | 176.45 |
| | 20 | 246.34 | 228.88 | 123.36 | 240.54 |
| | 30 | 290.78 | 288.18 | 157.93 | 293.76 |
| | 40 | 331.74 | 347.48 | 192.51 | 340.83 |
| Granodiorite | 0 | 164.62 | 174.11 | 64.80 | 164.62 |
| | 1 | 176.79 | 183.30 | 68.83 | 177.90 |
| | 2.5 | 203.88 | 197.09 | 74.87 | 196.37 |
| | 5 | 217.59 | 220.06 | 84.94 | 224.25 |
| | 10 | 289.88 | 266.01 | 105.08 | 272.76 |
| | 15 | 310.43 | 311.96 | 125.21 | 315.03 |
| | 20 | 347.14 | 357.91 | 145.35 | 353.15 |

To further analyze the error between the calculation results of the M–C, D–P, and E–D criteria and the test results, Equation (13) was used to calculate the average relative error and root mean square error between the theoretical and test values [25]. The results are shown in Figure 11.

$$
\begin{cases}
MRE = \dfrac{1}{N} \sum_{i=1}^{N} \dfrac{\left| \sigma_{1i}^{test} - \sigma_{1i}^{calc} \right|}{\sigma_{1i}^{test}} \times 100\% \\[4mm]
RM = \sqrt{\dfrac{\sum_{i=1}^{N} \left( \sigma_{1i}^{test} - \sigma_{1i}^{calc} \right)^2}{N}}
\end{cases}
\tag{13}
$$

where *MRE* is the average relative error, *RM* is root mean square error, *N* is the number of triaxial test groups, $\sigma_{1i}^{test}$ is the test value, and $\sigma_{1i}^{calc}$ is the calculated value.

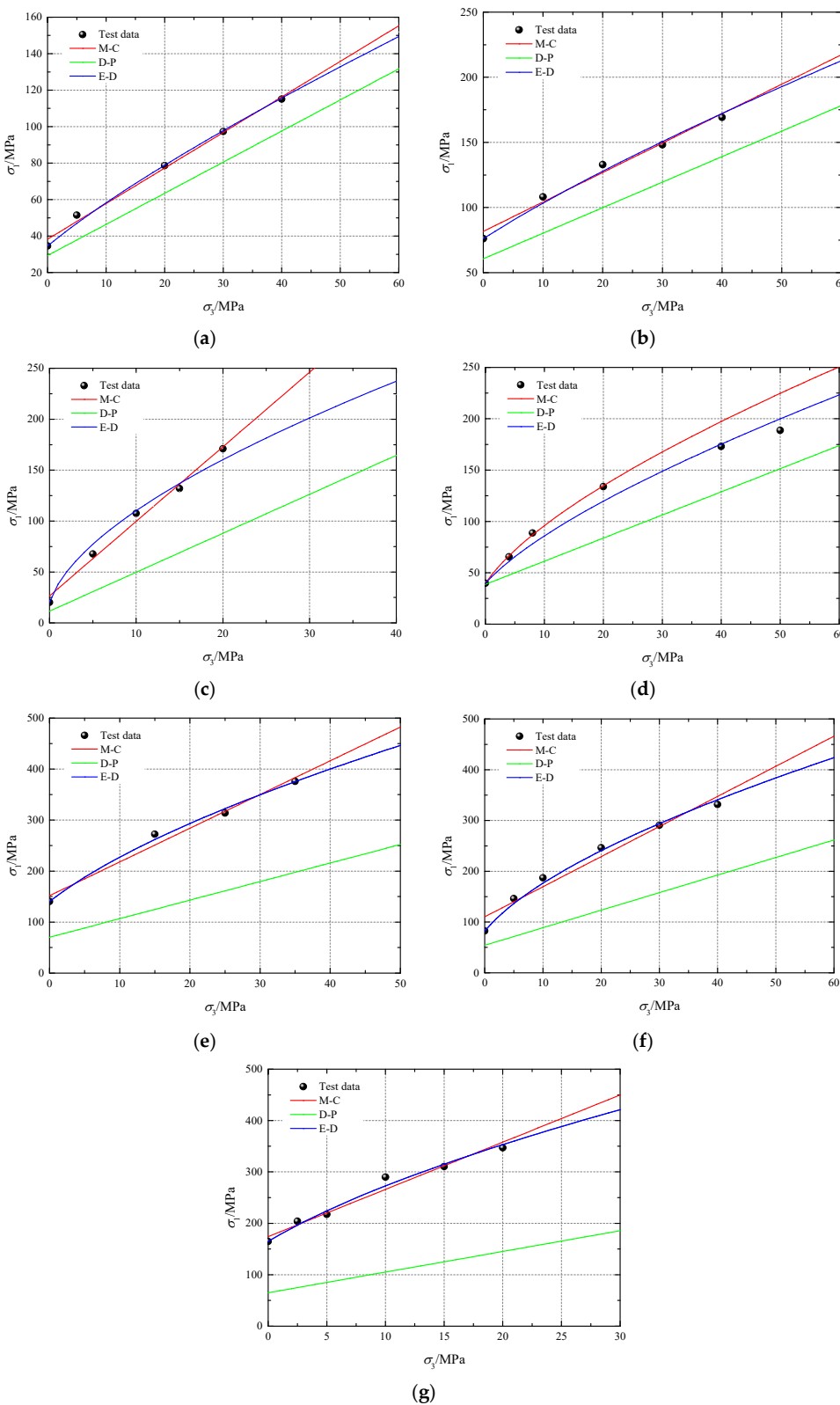

**Figure 10.** Comparison of experimental and calculated values for seven types of rocks. (**a**) Chlorite schist. (**b**) Hengda sandstone. (**c**) Argillaceous dolomite. (**d**) Jinping greenschist. (**e**) Huashan granite. (**f**) Jinping marble. (**g**) Granodiorite.

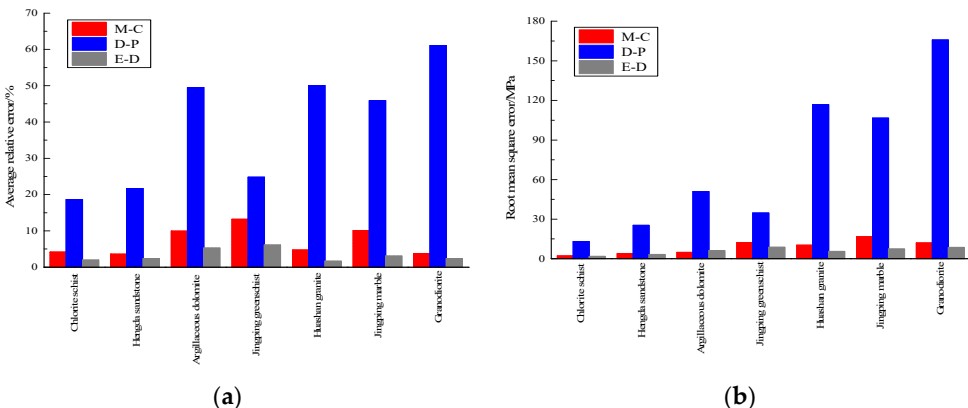

**Figure 11.** Error analysis: (**a**) the average relative error; (**b**) the root mean square error.

As shown in Figure 11, the average relative errors of the seven types of rocks calculated using the energy density criterion were 1.99%, 2.34%, 5.30%, 6.11%, 1.69%, 3.10%, and 2.36%, and the root mean square errors were 1.88, 3.28, 6.28, 8.80, 5.53, 7.50, and 8.68 MPa, respectively. Except for muddy dolomite, the average relative errors and root mean square errors of the other six types of rocks were the smallest, indicating that the E–D criterion had a higher accuracy than the M–C criterion and D–P criterion. The error analysis further demonstrates the applicability and accuracy of the rock energy density failure criterion.

## 6. Conclusions

(1) In the conventional triaxial compression deformation process, the input energy density of sandstone continued to increase, while the elastic strain energy density gradually increased in the pre-peak stage. When the energy storage limit of sandstone was exceeded, the sandstone was damaged, the elastic strain energy density decreased sharply, and the dissipative energy density increased rapidly.

(2) The peak elastic strain energy density increased linearly with the confining pressure; based on this, the rock energy density failure criterion under conventional triaxial compression was established. This criterion is simple in form, contains only a few parameters, and the physical meaning of each parameter is clear. The energy density failure criterion can be transformed into the Hoek–Brown failure criterion through parameter transformation, indicating that the Hoek–Brown failure criterion is a special case of the energy density failure criterion.

(3) The Mohr–Coulomb, Drucker–Prager, and energy–density criteria were used to calculate the strengths of the seven types of rocks under different confining pressures, and the energy–density criterion was the closest to the experimental results, indicating that the feasibility and accuracy of the energy–density criterion were higher than those of the Mohr–Coulomb and Drucker–Prager criteria.

**Author Contributions:** Conceptualization, L.Z.; methodology and validation, Y.C.; formal analysis, L.Z. All authors have read and agreed to the published version of the manuscript.

**Funding:** This research was funded by the Natural Science Research Project in the Universities of Anhui Province, grant number KJ2021A1056.

**Institutional Review Board Statement:** Not applicable.

**Informed Consent Statement:** Not applicable.

**Data Availability Statement:** The data used to support the findings of this study are available from the corresponding author upon request.

**Conflicts of Interest:** The authors declare no conflict of interest.

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
