# Peer review of "Study on Rock Failure Criterion Based on Elastic Strain Energy Density"

_applsci, doi:10.3390/app13148435_

Round 1
Reviewer 1 Report
Dear authors, Check out the following points
1. Rewrite the abstract to reflect the results' value addition.
2. The authors provided a nomenclature list for the interest of all readers.
3. This study framework should be included.
4. Should add highlight uniqueness of this paper.
5. Check journal format page 10, lines 259-262.
6. Authors could include additional description because all the explanations are about mechanical stresses with no emphasis on rock formation and geological setting.
7. Check journal format page 13, lines 302-306.
8. The conclusion should have more specific.
Regards

Author Response
Point by point responses to reviewers:
First, we would like to thank the reviewers and the editor for the positive and constructive comments and suggestions:
- Rewrite the abstract to reflect the results' value addition.
Answer: Thank you for your comments. The abstract has been rewritten to highlight valuable research results.
- The authors provided a nomenclature list for the interest of all readers.
Answer: Thank you for your comments. The author used a nomenclature list to introduce the current research status at home and abroad, in order to attract the interest of all readers.
- This study framework should be included.
Answer: Thank you for your comments. In the third paragraph of the Introduction in the revised manuscript, the author reorganized the study framework. Please review it.
- Should add highlight uniqueness of this paper.
Answer: Thank you for your comments. The author has added highlights and uniqueness of this paper in the Abstract, Introduction, and Conclusion sections, and highlighted them in red font. Please review it.
- Check journal format page 10, lines 259-262.
Answer: Thank you for your comments. Page 10, lines 259-262, have been modified according to the journal format requirements.
- Authors could include additional description because all the explanations are about mechanical stresses with no emphasis on rock formation and geological setting.
Answer: Thank you for your comments. The author introduced the acquisition of rock formation and geological setting for triaxial testing. Sandstone was taken from the roof bedrock of No. 8 coal seam in Banji Coal Mine, Lixin County, Anhui Province, China, with a buried depth of 619m~652m.
- Check journal format page 13, lines 302-306.
Answer: Thank you for your comments. Page 13, lines 302-306, have been modified according to the journal format requirements.
- The conclusion should have more specific.
Answer: Thank you for your comments. The conclusions have been reorganized and written according to the requirements of the reviewer in the revised manuscript, please review it.

Reviewer 2 Report
It is an interesting and valuable research work and it will be recommended for publication subject to a satisfactory updating/correction:
1. update the current state of art of research in this field, and clearly report what has been done and what is the new in this paper?
2. I noted that the missing of some more recent publication in this field.
3. justification of the experimental setting, accuracy, and repeatability;
4. improve the clarity of the wording or sentence, such as: "the energy density failure criterion of sandstone exhibiting high precision and few parameters with distinct physical meaning was established." what do you mean by established?
5. Conclusion:
1. should clearly distinguish what originally new from your work or confirmation of an existing general conclusion;
2. the context for each conclusion;
Author Response
Point by point responses to reviewers:
First, we would like to thank the reviewers and the editor for the positive and constructive comments and suggestions:
- update the current state of art of research in this field, and clearly report what has been done and what is the new in this paper?
Answer: Thank you for your comments. The Introduction of the revised manuscript has updated the current research state of rock energy evolution laws and failure criteria, and then clearly reported the research work done in this paper, as well as some new findings made. Please review it.
- I noted that the missing of some more recent publication in this field.
Answer: Thank you for your comments. Some more recent publications have been added to the Introduction in the revised manuscript. Please review it.
- justification of the experimental setting, accuracy, and repeatability.
Answer: Thank you for your comments. The size information of sandstone samples has been added in the revised manuscript, and the conventional triaxial compression test is a basic rock mechanics test, and its test methods and procedures have been very mature. Moreover, the test device, loading procedure and loading rate of axial and confining pressure are given in the paper, which is sufficient to demonstrate the accuracy and repeatability of the test.
- improve the clarity of the wording or sentence, such as: "the energy density failure criterion of sandstone exhibiting high precision and few parameters with distinct physical meaning was established." what do you mean by established?
Answer: Thank you for your comments. The sentence " the energy density failure criterion of sandstone exhibiting high precision and few parameters with distinct physical meaning was established." has been improved in the revised manuscript, and the revised result is: the energy density failure criterion of sandstone was established, and the criterion has high precision and few parameters, and the parameters have clear physical meaning.
- Conclusion: 1. should clearly distinguish what originally new from your work or confirmation of an existing general conclusion; 2. the context for each conclusion
Answer: Thank you for your comments. The conclusions have been reorganized and written according to the requirements of the reviewer in the revised manuscript, please review it.

Reviewer 3 Report
According to the linear growth relationship between the peak elastic strain energy density of sandstone and the confining pressure, the energy density failure criterion is established in this study. This topic is interesting and some experimental studies have been carried out. Some useful conclusions and relevant creep mechanism have been obtained. However, there are still some problems that require the author to explain and make minor modifications, as shown below:
(1) The author assumes that the volumetric strain during rock failure is 0, why can Equation (10) be transformed into Equation (11)?
(2) What are the advantages of the energy density failure criterion compared to the Hoek–Brown failure criterion?
(3) Figures (a) and (b) in Figure 14 do not have a title, please add.
(4) Some of the variables (c, φ) in the paper are not explained, please add.
(5) Please add the size information of the concrete standard sample
Minor revisison
Author Response
Point by point responses to reviewers:
First, we would like to thank the reviewers and the editor for the positive and constructive comments and suggestions:
- The author assumes that the volumetric strain during rock failure is 0, why can Equation (10) be transformed into Equation (11)?
Answer: Thank you for your comments. When the volumetric strain is 0, there is:
(1)
During the conventional triaxial compression test, , therefore, =0.5. Substituting =0.5 into Equation (10) to obtain Equation (11).
- What are the advantages of the energy density failure criterion compared to the Hoek–Brown failure criterion?
Answer: Thank you for your comments. Compared to the classic Hoek-Brown failure criterion, the advantage of the energy density failure criterion is that its use is not limited by the third principal stress. However, the third principal stress of the Hoek-Brown failure criterion should be less than half of the uniaxial compressive strength.
- Figures (a) and (b) in Figure 14 do not have a title, please add.
Answer: Thank you for your comments. The title of Figures (a) and (b) in Figure 14 has been added to the revised manuscript. Please review it.
- Some of the variables (c, φ) in the paper are not explained, please add.
Answer: Thank you for your comments. The definitions of the variables (c, φ) have been added to the revised manuscript. Please review it.
- Please add the size information of the concrete standard sample.
Answer: Thank you for your comments. The size information of the sandstone sample has been added to the revised manuscript. Please review it.

Round 2
Reviewer 1 Report
Thank you for your efforts to up date the final version